# Defense Mechanism to Generate IPS Rules from Honeypot Logs and Its Application to Log4Shell Attack and Its Variants

**Yudai Yamamoto * and Shingo Yamaguchi ***

Graduate School of Sciences and Technology for Innovation, Yamaguchi University, 2-16-1 Tokiwadai,
Ube 755-8611, Japan
* Correspondence: d011wcu@yamaguchi-u.ac.jp (Y.Y.); shingo@yamaguchi-u.ac.jp (S.Y.)

**Abstract:** The vulnerability of Apache Log4j, Log4Shell, is known for its widespread impact; many attacks that exploit Log4Shell use obfuscated attack patterns, and Log4Shell has revealed the importance of addressing such variants. However, there is no research which focuses on the response to variants. In this paper, we propose a defense system that can protect against variants as well as known attacks. The proposed defense system can be divided into three parts: honeypots, machine learning, and rule generation. Honeypots are used to collect data, which can be used to obtain information about the latest attacks. In machine learning, the data collected by honeypots are used to determine whether it is an attack or not. It generates rules that can be applied to an IPS (Intrusion Prevention System) to block access that is determined to be an attack. To investigate the effectiveness of this system, an experiment was conducted using test data collected by honeypots, with the conventional method using Suricata, an IPS, as a comparison. Experimental results show that the discrimination performance of the proposed method against variant attacks is about 50% higher than that of the conventional method, indicating that the proposed method is an effective method against variant attacks.

**Keywords:** IDS; IPS; variants; Log4Shell; defense system; honeypots; automaton

## 1. Introduction

Log4Shell, a vulnerability of the logging library Apache Log4j discovered in December 2021, had a wide impact given the library's use in many services and applications. The first attack exploiting Log4Shell was observed on 10 December 2021 [1]; the vulnerability in Log4Shell was registered with CVE in November [2], but was not publicly disclosed until 9 December [3], meaning that a widespread attack began within one day of the vulnerability's disclosure. The rulesets applicable to IPS and IDS on that day added rules for attacks that exploit Log4Shell [4], but they did not address the obfuscated attack pattern. Although there is no information on the Web regarding the timing of the appearance of obfuscated attack patterns, accesses using obfuscated attack patterns were observed on the following 11th in the data collection environment we ourselves set up, which will be described later. Thus, the response of the attacker to countermeasures taken by defenders such as IPS and IDS is quick, and as can be seen from the fact that obfuscated attack patterns began to appear one day after the appearance of non-obfuscated attack patterns, speed is very important in responding to variants. Thus, Log4Shell once again demonstrated the importance of defending against attacks using variants.

However, the defense methods proposed so far do not consider protection against variants. In this paper, we propose a defense system that targets not only known attacks, but also variant attacks. The proposed defense system can be divided into three main parts: honeypots, machine learning, and rules generation. By collecting data using honeypots, we can obtain information about the latest attacks that are currently being observed, enabling us to defend against the latest attacks. Machine learning is used to discriminate between

attacks and non-attacks on the data collected using honeypots. Rules are then generated to be applied to IPS to block accesses that are determined to be attacks as a result of the discrimination using machine learning. Finally, the generated rules are applied to the IPS to achieve protection.

We focus on Log4Shell not only because it presents a particularly high impact vulnerability, as mentioned above, but also because we intentionally limit our discussion in this paper to focus on known attacks and their variants.

To investigate the effectiveness of the proposed defense system, experiments were conducted to compare the effectiveness of the proposed defense system against known and variant attacks, using Suricata with the ET ruleset [5,6], the conventional method, as the comparison target. The data used in the experiments were actually data collected using honeypots. Machine learning results showed that for known attacks, the proposed method has approximately 90% higher discrimination performance than the conventional method. For variant attacks, the proposed method shows a discrimination performance approximately 50% higher than the conventional method. In addition, a comparison of generated patterns for attacks that exploit Log4Shell shows that the proposed method is 40% better than the conventional method using Snort with the Snort Community ruleset [7,8] in terms of the understandability of the patterns, although the discrimination performance against the attacks is the same. These results show that the proposed method is more effective than the conventional method not only against known attacks but also against variant attacks.

The three major contributions of this paper are listed below.

1.  After clarifying that none of the methods proposed so far enable autonomous defense systems or use black-box methods and that no variant-aware pattern generation method exists, we proposed a defense system that satisfies all of them.
2.  The Suricata ET ruleset was compared with the proposed method for attack detection using the conventional method, and the results show that the proposed method has approximately 90% higher discrimination performance than the conventional method for known attacks and approximately 50% higher discrimination performance than the conventional method for variant attacks, indicating that the proposed method has better discrimination performance than the conventional method for both known and variant attacks.
3.  The results of the comparison of the patterns generated by the proposed method with the Snort Community ruleset as the conventional method against attacks that exploit Log4Shell showed that the proposed method is superior to the conventional method in terms of the discrimination performance and the understandability of the patterns by 40%, indicating that the proposed method is superior to the conventional method in terms of the discrimination performance and the understandability.

In Section 2, we describe the works related to this research. In Section 3, we introduce the proposed defense system and describe its components. In Section 4, we describe the experiments conducted to investigate the effectiveness of the proposed defense system and their results. In Section 5, we discuss the results obtained from the experiments. In Section 6, we present our conclusions.

## 2. Related Works

### 2.1. Autonomous Defense System

In our previous method [9], we proposed an autonomous defense system and showed its effectiveness against both known attacks and variant attacks, but focused on attack detection and not rules generation. In this paper, we aim to target not only the detection of attacks, but also the generation of rules.

The intrusion detection method proposed by *Umer* et al. [10] showed higher effectiveness than machine learning methods such as SVM due to deep-learning-based classification. However, it only detects attacks and does not focus on protecting against them.

The honeypot system [11] proposed by *Pa* et al. combines low-interaction and high-interaction honeypots and uses the results of the interaction between the high-interaction honeypot and the attacker to strengthen the low-interaction honeypot. It differs from this research in that it targets only IoT devices and uses autonomy to strengthen honeypots rather than defend against attacks.

The IDS proposed by *Awajan* has been shown to be more than 90% effective and performs as highly as an IDS [12], but differs from this research in that it targets IoT devices and is not a pattern-based IDS, making it less versatile.

*Jiang* et al. proposed a honeypot that targets attacks on the Web and is augmented by machine learning [13]. It aims to reduce the effort in data analysis by solving issues such as noise and wasted data when analyzing the data collected by honeypots. Machine learning is used in data analysis and is not used to determine whether an attack is genuine.

*Ghourabi* et al. proposed a system for detecting attacks on web services by capturing communications between a honeypot that simulates a web service and an attacker, and performing machine learning on the data [14]. Although they share the same use of machine learning to determine whether an attack is genuine, they use a learning environment to collect data that is not an attack, are not complete within the defense system, and do not focus on protecting against the latest attacks by using the most recent data.

IPS and IDS are both widely used security concepts; the difference between IPS and IDS is that IPS has the ability to automatically block access to an attack if it is detected, while IDS only detects attacks. Most IPS and IDS are signature-based. Signature-based IPS and IDS are enabled by applying rules that specify what kind of accesses are targeted. The rules applied to IPS and IDS are created by humans and are not automatically generated.

### 2.2. Pattern Generation Methods for Application to IPS and IDS

*Wang* et al. proposed a method to obtain information about Log4Shell from information on Twitter [15]. In their own research, they utilize information from the Web in pattern generation, which is a similar approach. However, [15] focuses on obtaining information and does not focus on utilizing the information obtained.

The Snort signature generation technique using natural language processing techniques proposed by *Laryea* employs a relatively new large-scale language model called GPT-2 [16], but it is not clear whether it can generate highly effective rules for a completely different set of attacks, as the technique generates new signatures based on existing signatures.

The Snort rule generation method proposed by *Jaw* et al. [17] does not use black-box methods such as machine learning, but focuses only on individual accesses and does not consider the variant perspective.

Thus, there are still no prior studies that use machine learning for discriminating attacks to enable autonomous defense systems or to white-box the process of dealing with variants and generating patterns. In addition, none of the previous studies listed above have considered the response to variant attacks. A comparison of the related studies and the defense system proposed in this paper is shown in Table 1.

**Table 1.** Comparison of related works and our proposed method.

| Method | Pa et al. [11] | Jiang et al. [13] | Ghourabi et al. [14] | Awajan [12] | Our Proposed Method |
|---|---|---|---|---|---|
| Automated system | ✓ | | | ✓ | ✓ |
| Using machine learning | | ✓ | ✓ | ✓ | ✓ |
| Detecting attacks | ✓ | | ✓ | ✓ | ✓ |
| Defending against attacks | | | ✓ | ✓ | ✓ |
| Targeting Web applications | | ✓ | ✓ | | ✓ |
| Consideration of variants | | | | | ✓ |

The important elements of the defense system proposed in this paper are machine learning to identify attacks and rule generation to generate rules that can be applied to IPS and IDS, and the performance of each was compared with that of conventional methods in experiments.

### 2.3. Log4Shell

Log4Shell is a vulnerability of Apache Log4j discovered in December 2021. Because of its use in various applications and services and the ease of exploiting the vulnerability, Log4Shell became a vulnerability that affected a wide range of applications [18]. It is possible to conduct an attack simply by preparing a pattern to be used in the attack and having Log4j process the string in some way. Finally, by making Log4j load Java classes, an attacker can make Log4j execute arbitrary commands or programs [19].

Figure 1 shows the variations of Log4Shell attack patterns [18,20].

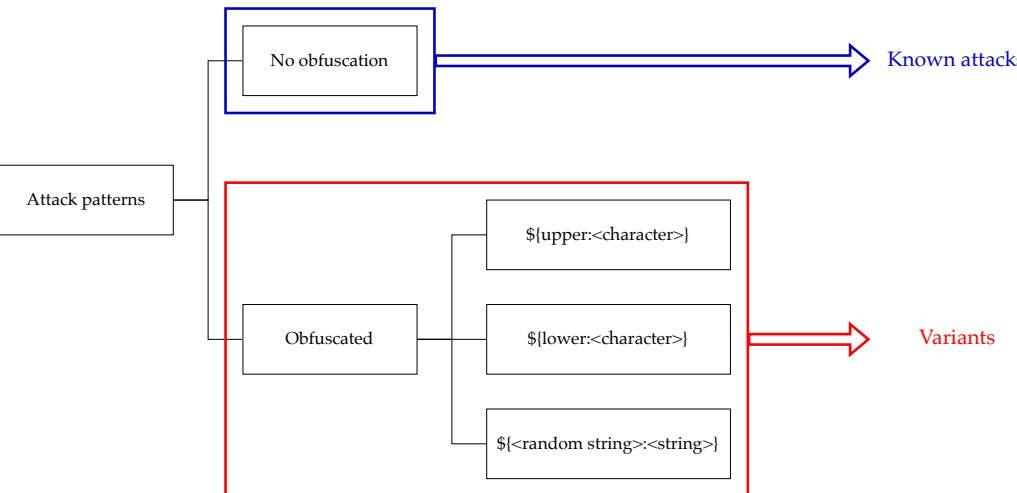

**Figure 1.** Variations of attack patterns exploiting Log4Shell.

*Hiesgen* et al. observed and analyzed attacks that exploit Log4Shell and found that attacks that exploit Log4Shell specifically use the LDAP protocol, and that attacks that exploit Log4Shell have been observed even after the release of Log4j, which fixes the vulnerability [21]. These attacks have been observed even after the release of Log4j, which fixes the vulnerability. There are also honeypots that specialize in observing attacks that exploit Log4Shell, such as `Log4Pot` [22] and `log4j-honeypot-flask` [23].

As mentioned above, the technique proposed by *Kaushik* et al. is used not only to attack but also to prevent attacks that exploit Log4Shell, due to the low difficulty of executing such attacks [24]. In addition, the method proposed by *Xiao* et al. focuses on communication with LDAP servers and achieves protection against attacks that exploit Log4Shell [25]. However, the method of *Kaushik* et al. [24] is effective only against attacks that exploit Log4Shell, and is not an effective technique against Log4j vulnerabilities or non-Log4j attacks other than Log4Shell. Also, the method of *Xiao* et al. [25] is not effective against them because, as *Hiesgen* et al. [21] clarified, attacks that exploit Log4Shell may use protocols other than LDAP.

These related works on Log4Shell are categorized as shown in Figure 2.

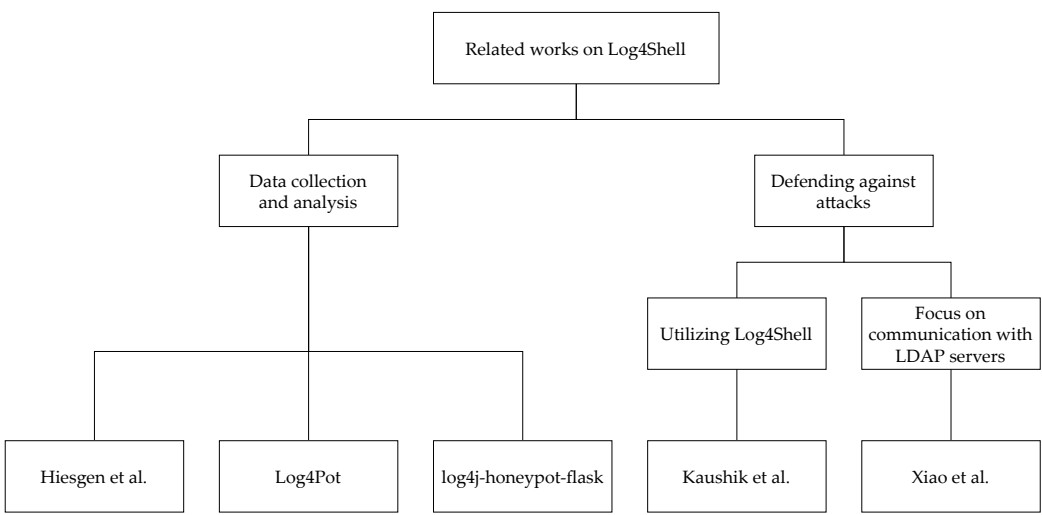

**Figure 2.** Dendrogram of works related to Log4Shell. Hiesgen et al. references [21], Log4pot references [22], log4j-honeypot-flask references [23], Kaushik et al. references [24] and Xiao et al. references [25].

### 3. Defense System Combining Honeypots and IPS

*3.1. Overview*

In order to defend against both known attacks and variant attacks, we propose the defense system shown in Figure 3.

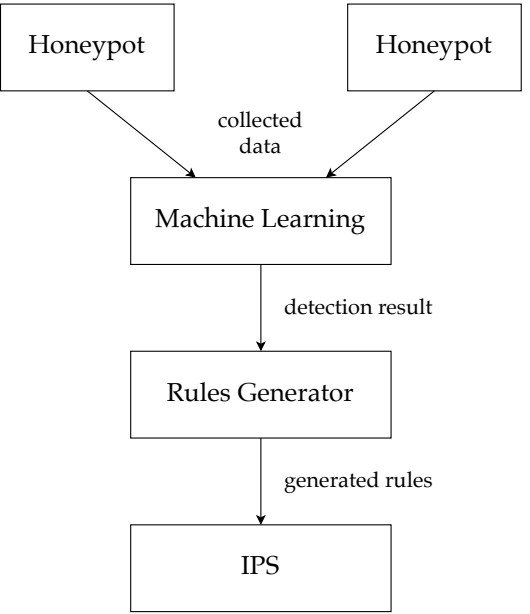

**Figure 3.** Diagram of the proposed defense system.

*3.2. Data Collection Environment Using Honeypots*

In this research, we use two environments for data collection: one using WOWHoney-pot [26] and the other installing WordPress [27], which is widely used for blog sites.

First, the characteristics of the data collection environment using WOWHoneypot are listed below.

- The operating system to be used is Alpine Linux [28], which is known as a lightweight Linux operating system.
- In addition to port 8080, which WOWHoneypot initially accepts access to, it also supports access to port 80, which is used as the HTTP port.

- The WOWHoneypot settings are not changed, and the observation is performed with the default settings.

Next, the characteristics of the data collection environment with WordPress installed are listed below.

- The OS we use is Ubuntu Server 22.04 LTS.
- Apache2, MySQL, and PHP are used to run WordPress.
- All PHP extensions and packages required to run all WordPress functionality are installed [29].
- To prevent the effects of vulnerabilities in WordPress, it is always updated to the latest version.
- Accepts access via HTTP and HTTPS.

All observed accesses are recorded in the access log; in the WordPress execution environment, the Apache2 web server records the observed accesses in the access log.

Next, we describe the geographic locations indicated by the IP addresses assigned to each environment. The WordPress environment built on DigitalOcean [30] is assigned an IP address in the US, while the WOWHoneypot environment uses an LTE line with a fixed IP address in Japan. The different geographical locations indicated by the IP addresses of each data collection environment are expected to enable the collection of data with different trends in each data collection environment.

### 3.3. Machine Learning Using RapidMiner

In this research, machine learning is performed using RapidMiner [31]. RapidMiner is software that can be used for machine learning, data mining, etc. RapidMiner has the concept of "operators" and "processes". An operator is an operation to realize a desired process, and a collection of operators (one or more) is called a process. In each of the following processes, the default values for each of the RapidMiner operators used are used as parameters.

The flow of discrimination by machine learning is shown below.

1. Loading of the learning data
2. Preprocessing of the learning data
3. Learning the model (machine learning)
4. Loading of the test data
5. Preprocessing of the test data
6. Validate the model (performance evaluation)

This section describes the training data and the test data. Both data consist of two columns: a URL string and a label. To focus on the "words" in the URL string, remove all but the first "/".

**before removing:** `/sqlite/SQLiteManager-1.2.4/main.php`

**after removing:** `/sqlite SQLiteManager-1.2.4 main.php`

Each URL string is labeled either "attack" to indicate that it is an attack or "clean" to indicate that it is not an attack. The conditions for the "attack" label are listed below.

- This is a URL string that has been known as an attack to exploit the vulnerability.
- This is not a vulnerability, but it is a URL string that could be exploited as an attack.
- URL strings that attempt to access directories that are not supposed to be public directories (e.g., `.env`, `.git`, etc. (including strings beginning with "`.`" (dot)).
- URL strings that contain OS commands such as `/bin/sh` and have a high possibility of OS command injection.

### 3.3.1. Loading of the Training Data

Load the training data in CSV format. The training data consist of URL strings and labels, as described above; the URL strings are treated as text, and the labels are binalized

since there are only two types of labels, "attack" and "clean", as described above. The process of implementing these processes in RapidMiner is shown in Figure 4.

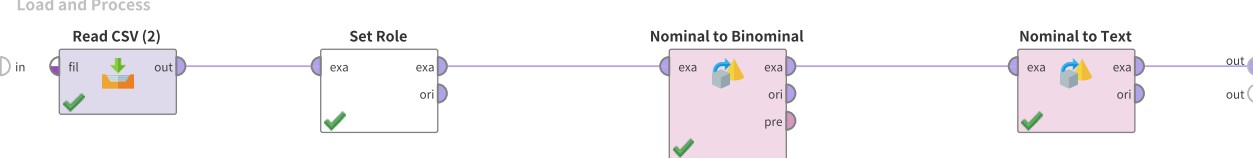

**Figure 4.** Implementation of the training data loading process in RapidMiner.

The process shown in Figure 4 uses the "Read CSV" operator to read the training data, and subsequent operators are used to treat the URL string as text and to binarize the labels.

### 3.3.2. Preprocessing of the Training Data

To make the training data usable as training data for machine learning, the URL string is vectorized. The delimiter character for vectorization is a one-byte space. The vectorization process performed on the training data is also performed on the test data. The process of implementing this process in RapidMiner is shown in Figure 5.

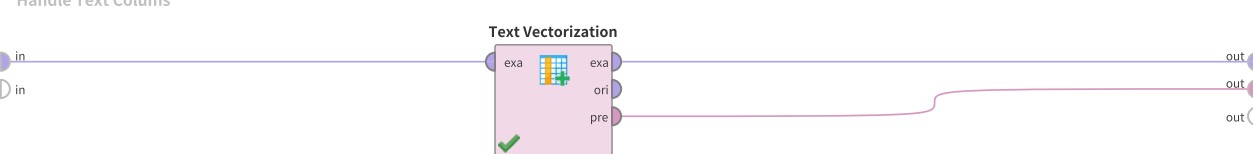

**Figure 5.** Implementation of the training data preprocessing in RapidMiner.

The process shown in Figure 5 uses the `Text Vectorization` operator to vectorize the URL strings in the training data. The `Text Vectorization` operator allows the same process to be used when applying the same vectorization to different data. The output `pre` of the `Text Vectorization` operator is used during the preprocessing of the test data to perform the same processing that was performed on the training data.

### 3.3.3. Machine Learning of the Model

In training the model, machine learning is performed using the training data described above. The following nine machine learning methods are used. The discriminant performance of each method is compared, and the best method is selectively used.

- Naive Bayes
- Generalized Linear Model
- Logistic Regression
- Fast Large Margin
- Deep Learning
- Decision Tree
- Random Forest
- Gradient Boosted Trees
- Support Vector Machine

Implementation of the above process in RapidMiner is shown in Figure 6.

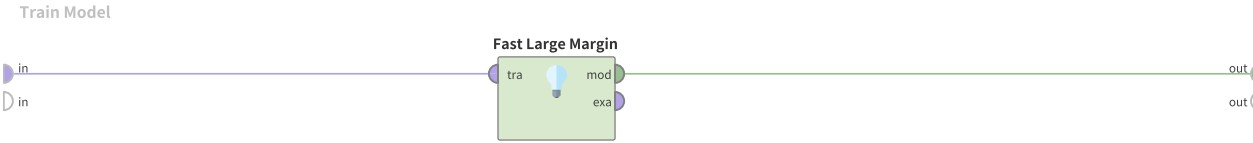

**Figure 6.** Implementation of the model training in RapidMiner.

The process shown in Figure 6 shows the process when the Fast Large Margin is used, which can be modified to use another machine learning method.

### 3.3.4. Loading of the Test Data

The test data is a CSV file with the same data structure as the training data, and the process performed when loading the test data is the same as the process performed when loading the training data: the URL string is treated as text and the label is binary since there are only two types of labels, "attack" and "clean". The process of implementing these processes in RapidMiner is shown in Figure 7.

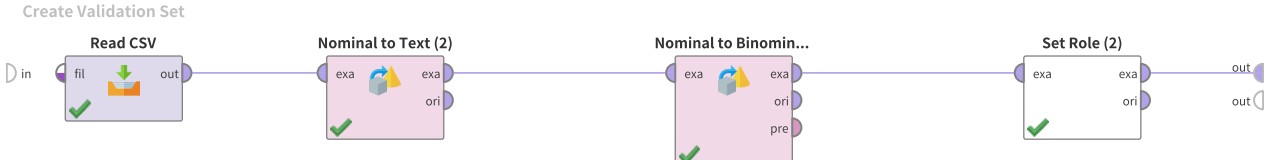

**Figure 7.** Implementation of the test data loading process in RapidMiner.

The process shown in Figure 7 reads the test data using the "Read CSV" operator. Subsequent operators are performed to convert the test data type in the same way as was done for the training data and do not affect the data content.

### 3.3.5. Preprocessing of the Test Data

To make the test data applicable to the model generated by machine learning, the URL string is vectorized. The vectorization process is the same as the vectorization performed on the training data. The process of implementing this process in RapidMiner is shown in Figure 8 .

In Figure 8, the `Apply Model` operator, named "Apply TV", is used to perform the same vectorization process that was applied to the training data.

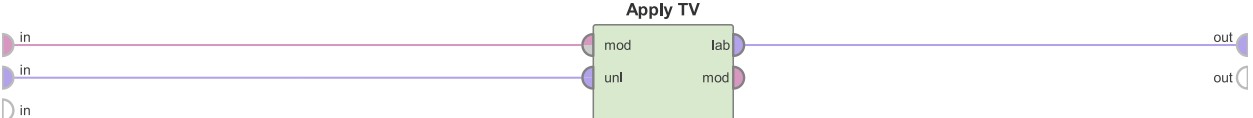

**Figure 8.** Implementation of the test data preprocessing in RapidMiner.

### 3.3.6. Evaluation of the Model Using the Test Data

In the model evaluation, the machine-learning-generated model is applied to the test data that have undergone the aforementioned preprocessing and produces a confusion matrix that is used to determine how many URL strings were correctly discriminated. The process of implementing these processes in RapidMiner is shown in Figure 9.

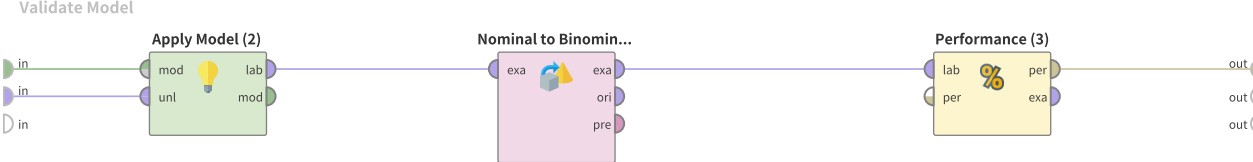

**Figure 9.** Implementation of the model evaluation using the test data in RapidMiner.

In the process shown in Figure 9, the `Apply Model` operator is used to apply the model to the test data, and the operator "Performance" is used to obtain the confusion matrix. The operators in between them perform the necessary type transformations to obtain the confusion matrix and do not affect the final output.

### 3.4. Generation of Rules Applicable to IPS and IDS

The flow of rules generation using the method of rules generation using prior knowledge is shown in Figure 10.

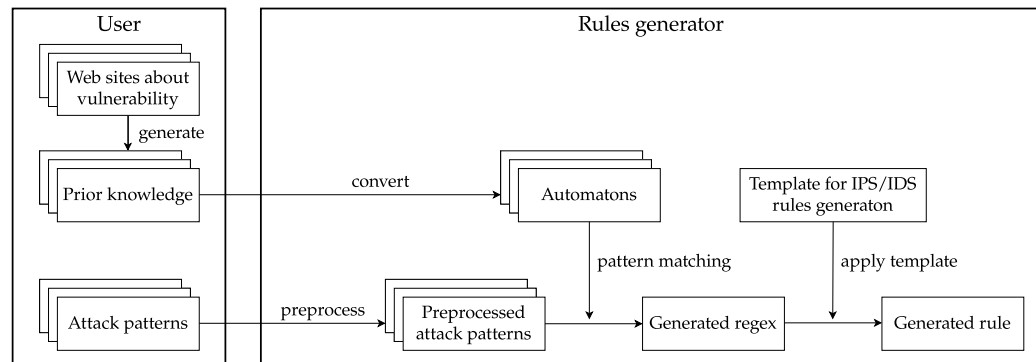

**Figure 10.** The flow of rules generation using prior knowledge.

We generate rules to be applied to IPS to block accesses using URL strings identified as attacks by the machine learning. The process of generating rules is shown in Algorithm 1.

---

**Algorithm 1** Generate rules to be applied to IPS.

---

The user registers the obfuscation pattern in the dictionary of obfuscation methods in advance
Convert obfuscation methods to automatons
**Require:** *A*: Array of prior knowledge converted to automatons, *U*: Array of URL strings given as input
**Ensure:** *R*: Generated rule
    **for all** *url* ∈ *U* **do**
        **for all** *automaton* ∈ *A* **do**
            Perform matching with *automaton*
            Generate pattern matching results
        **end for**
        Based on the pattern matching results, generate a regex pattern *R*
    **end for**
    Apply the template for IPS/IDS rules generation to *R*
    **return** *R*

---

To achieve the above process, a Java program was created to generate regex patterns and illustrate the generated automaton. The program consists of the six methods shown below and uses dk.brics.automata [32] as a library to handle regexes and automatons.

1.   `read` (Read the file containing the string to be processed)
2.   `prepare` (Remove the first character (sequence) and the last character (sequence) that must appear from the string to be processed)
3.   `convert` (Generate finite automaton from regex patterns)
4.   `match` (Perform matching using finite automaton)
5.   `generate` (Generate regex patterns and finite automaton based on matching results)
6.   `check` (Verify that the generated regex actually matches correctly)

#### 3.4.1. Reading Files to Process

The `read` method reads strings one by one from a file containing strings that match the patterns and stores them in an array. The file name is taken as input, and the array consisting of the read strings is output.

Next, the `prepare` method removes the character (sequence) that always appears at first and the character (sequence) that always appears at last from each of the strings given as input. The reason for this process is to reduce the cost incurred in processing parts that are always obvious. The first and last strings, whether symbols, alphanumeric characters,

or a combination of both, are unique strings and will not be modified in any way when converted to regexes. Therefore, it is possible to generate a regex corresponding to the originally given string by simply converting any other string into a regex and adding the first and last occurrences of the string to the beginning and end of the regex, respectively.

### 3.4.2. Generating Finite Automaton from Regex Patterns

The `convert` method generates a finite automaton from prior knowledge (of a regex) given in the form of a string. The string of the regex pattern given as prior knowledge is taken as input, and an array consisting of the finite automaton generated is output.

### 3.4.3. Matching Using Finite Automaton

The `match` method performs matching with a finite automaton for each string to match. It takes as input an array of strings to be matched and an array of automatons generated from pre-expressions and outputs an array consisting of the finite automaton matched and the number of times they were matched. The operation of the matching process is shown in Algorithm 2.

---

**Algorithm 2** Matching using finite automaton.

---

**Require:** *arrayList*: An array of URL strings preprocessed given as input, *list*: Array of prior knowledge
**Ensure:** *hashMap*: The result of matching against prior knowledge, *not_matched*: An array of characters not matched against prior knowledge

> **function** MATCH(*arrayList*, *list*)
> > **for all** $s \in arrayList$ **do**
> > > $loc \leftarrow 0$
> > > **while** $loc <$ size of $s$ **do**
> > > > $matched = false$
> > > > **for all** $exp \in list$ **do**
> > > > > Matching by automaton $exp$
> > > > > $l \leftarrow$ Index matched to automaton $exp$
> > > > > **if** Matched to automaton $exp$. **then**
> > > > > > $loc + = l$
> > > > > > Increase the number of matches to $exp$ in $hashMap$ by 1
> > > > > > $matched = true$
> > > > > > **break**
> > > > > **end if**
> > > > **end for**
> > > > **if** $matched == false$ **then**
> > > > > Add the $loc$th character of $s$ to *not_matched*
> > > > > $loc + = 1$
> > > > **end if**
> > > > Generate rules using $hashMap$ and *not_matched*
> > > **end while**
> > **end for**
> **end function**

---

### 3.4.4. Generate Regex Patterns and Rules Based on Matching Results

The `generate` method outputs a regex pattern illustrated by a finite automaton based on the matching results. The process of generating regex patterns and finite automatons depends on the number of matched automatons and the number of times each was matched. The process of rule generation is shown in Algorithm 3.

---

**Algorithm 3** Generate regex patterns and rules based on matching results.

---

**Require:** *hashMap*: The result of matching against prior knowledge, *not_matched*: An array of characters not matched against prior knowledge
**Ensure:** *builder*: A generated rule
  **function** Generate(*hashMap*, *not_matched*)
    *max*: Maximum number of matches with each prior knowledge obtained from *hashMap*
    *min*: Minimum number of matches with each prior knowledge obtained from *hashMap*
    *num*: Number of matched prior knowledge obtained from *hashMap*
    *output*: An array of regex patterns to be finally combined into one
    **for all** *entry* $\in$ *hashMap* **do**
      Add *entry* to *output*
    **end for**
    *symbols*: An array containing the symbols in *not_matched*
    **if** size of *not_matched* $>$ 1 **then**
      *regex*: An array containing regex patterns generated based on the characters contained in *not_matched*
      **if** *contains_alpha* $==$ *true* **then**
        Add `[a-zA-Z]` to *regex*.
      **end if**
      **if** *contains_digit* $==$ *true* **then**
        Add `[0-9]` to *regex*.
      **end if**
      **for all** *ch* $\in$ *symbols* **do**
        Add *ch* to *regex*.
      **end for**
      Add all items in *regex* to *output*
    **end if**
    *builder*: The final output regex pattern string
    **if** *num* $>$ 1 **then**
      **if** *max* $\geq$ 1 and *min* $\geq$ 1 **then**
        **for** $i = 0; i <$ size of *output*; $i + +$ **do**
          **if** $i == 0$ **then**
            Add ( to *builder*
          **end if**
          Add the *i*th item of *output* to *builder*
          **if** $i <$ size of *output* $- 1$ **then**
            Add | to *builder*
          **end if**
          **if** $i ==$ size of *output* $- 1$ **then**
            Add ) to *builder*
            **if** *max* $>$ 1 **then**
              Add + to *builder*.
            **end if**
          **end if**
        **end for**
      **end if**
    **end if**
    Apply the template to *builder* to make it applicable to IPS and IDS
    **return** *builder*
  **end function**

---

Note that in the third and fourth conditions shown above, the same pattern is generated in both cases. This is because if one tries to generate a pattern that strictly satisfies the third condition, the generated pattern will become more complex and the understandability of the generated pattern will decrease.

The final output is a regex pattern and its automaton representation. Regex patterns can be used as patterns for rules to be applied to IPS and IDS, and automaton representation can help humans understand the generated regex patterns. Since automaton is a graphical representation of the pattern matching process, it is an easy-to-understand visualization technique that does not require knowledge of regexes. The reason for using automaton for pattern matching is that they have the advantage of being able to accept URL strings that are a combination of multiple obfuscation methods. In generating regex patterns from pattern matching with automaton, emphasis is placed on not considering the order of appearance of patterns given as prior knowledge. For example, consider the case where there are patterns *A* and *B* as prior knowledge, and both strings that appear in order $A \rightarrow B$ and $B \rightarrow A$ are included in the URL string given as input. Taking into account the order in which the patterns appear, two patterns will be generated, one matching *AB* and the other matching *BA*. However, since the order of appearance of patterns can be ignored in the attack patterns used in the Log4Shell exploits targeted in this paper, unnecessary patterns will be generated if the order of appearance is considered as in the example above. In the case of pattern generation without considering the order of appearance, a pattern is generated that matches a string containing *A* or *B*, and a single pattern can correspond to both *AB* and *BA*. Not considering the order of occurrence when generating patterns is used as a form of prior knowledge.

Rules applicable to IPS and IDS are generated using the regex patterns generated by the aforementioned process. To convert the generated regex patterns into rules applicable to IPS and IDS, some characters are escaped and applied to a template.

```
alert tcp $EXTERNAL_NET any -> $HOME_NET $HTTP_PORTS (msg:''Log4Shell'';
flow:to_server,established; content:''${''; fast_pattern:only; http_uri;
pcre:''<regex>''; metadata:policy balanced-ips drop, policy connectivity-
ips drop, policy max-detect-ips drop, policy security-ips drop, ruleset
community, service http; classtype:attempted-user; sid:58724; rev:6;)
```

This template shown above is based on the content of the already existing Snort Community ruleset, with some modifications (e.g., deletion of content that is output to the log and parts that do not affect discrimination performance).

A specific example of regex pattern generation is shown below. In this case, the following prior knowledge is assumed to be applied. This prior knowledge is obtained from the behavior of the existing attack pattern generator program [33] and the analysis [4,34] regarding attacks that exploit Log4Shell.

- `$\{upper:([a-z0-9]|:|:|\. |/)\}` is used as one of the obfuscation methods.
- `$\{lower:([a-z0-9]|:|:|\. |/)\}` is used as one of the obfuscation methods.
- `$\{([a-z0-9]|:)+:\-([a-z0-9])+\}` is used as one of the obfuscation methods.
- If multiple obfuscation methods are used in combination, their order of appearance is not considered.

Consider the case of generating regex patterns from the following attack pattern.

```
${${lower:j}n${lower:d}i${lower::}l${lower:d}${upper:a}${lower:p}s${lower
::}${lower:/}${lower:/}${upper:1}${upper:2}${upper:7}${lower:.}${upper:0}${
upper:.}${lower:0}${lower:.}${upper:1}${lower::}${upper:1}${lower:3}${upper
:8}${lower:9}${upper:/}t${lower:e}s${upper:t}}
```

First, the regex patterns given as prior knowledge are converted to a finite automaton. In this case, the above three regexes are converted to automaton. Each regex pattern is converted into a finite automaton, and a graphical representation is shown in Figure 11.

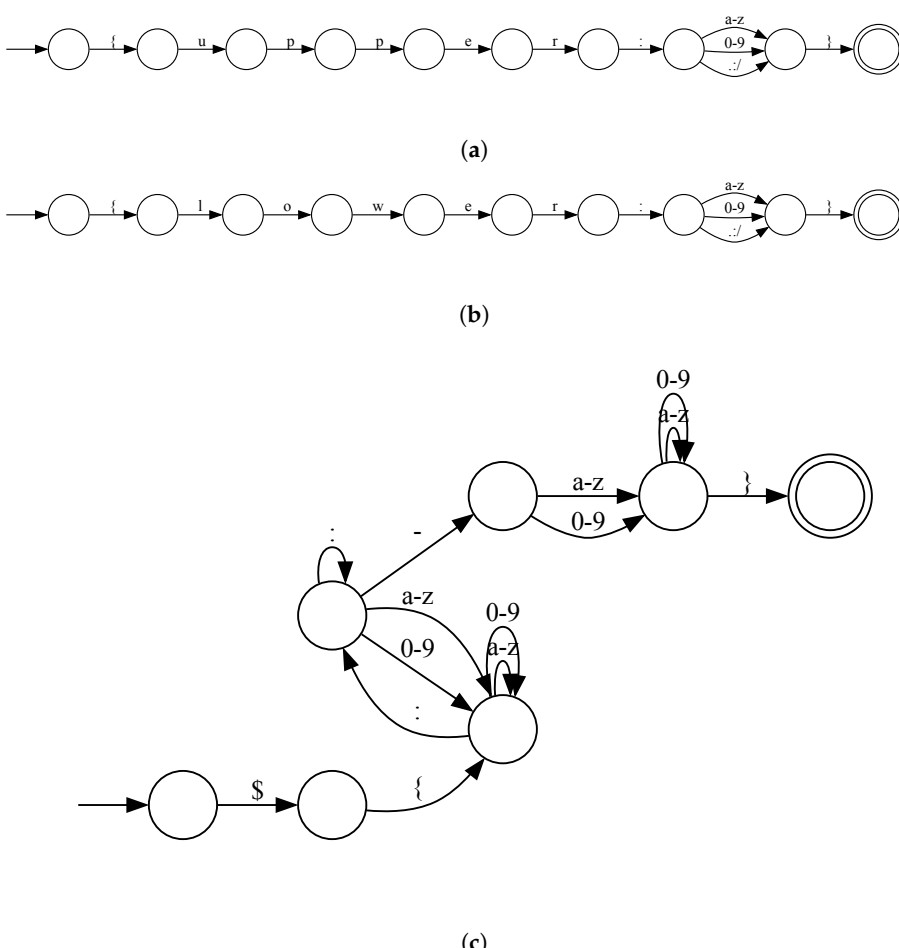

**Figure 11.** Visualization of the three regex patterns given as prior knowledge converted to finite automatons. (**a**) Visualization of `$\{upper:([a-z0-9]|:|\.|/)\}` converted to a finite automaton. (**b**) Visualization of `$\{lower:([a-z0-9]|:|\.|/)\}` converted to a finite automaton. (**c**) Visualization of `$\{([a-z0-9]|:)+:\-([a-z0-9])+\}` converted to a finite automaton.

Next, matching is performed using the automaton generated from the regex pattern. Finally, the regex pattern shown in Figure 12 is generated.

`($\{lower:([a-z0-9]|:|\.|/)\}|$\{upper:([a-z0-9]|:|\.|/)\}|[a-z])+`

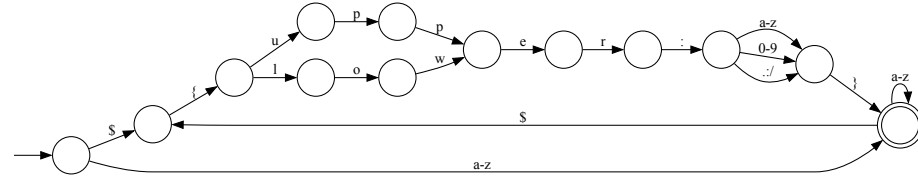

**Figure 12.** Result of visualization of the final generated finite automaton.

## 4. Confirmation of the Effectiveness of Defense System

### 4.1. Experimental Flow and Evaluation Metrics

To investigate the effectiveness of the proposed defense system, 4 experiments were conducted as described below.

**Experiment 1.** Extract variants from test data for validation.

**Experiment 2.**　Use Suricata with the ET ruleset applied to examine the effectiveness of conventional methods against known attacks and variants.

**Experiment 3.**　Use RapidMiner to examine the effectiveness of the proposed method against known attacks and variants.

**Experiment 4.**　Use a turn generation program to generate rules for attacks that exploit Log4Shell and compare the generated rules with those contained in the Snort Community ruleset.

The ET ruleset used in Experiment 2 was downloaded in January 2023. Although the ruleset available for download was divided into files by category, the experiment was conducted so that all files were read.

In Experiment 3, we used a machine with an Intel Xeon W-1270 CPU and 40 GB of memory for machine learning with Rapid Miner. It should be noted that machine learning using RapidMiner requires a lot of memory, so machines with less than 40 GB of memory may be affected. In addition, the machine learning methods shown in the experimental results are based on the selective use of one of the best-performing methods available in RapidMiner.

The Snort Community rule set used in Experiment 4 was downloaded in March 2023. It includes rules for other attacks as well as Log4Shell, but only rules related to Log4Shell were extracted. We extracted rules that contain "Log4j" or "Log4Shell" in the comment of the rule. The extraction resulted in 36 rules, but only one rule corresponding to all three obfuscation methods, and the non-obfuscated attack pattern shown in Figure 1 was used in the experiment.

The True Positive Rate (TPR) and the True Negative Rate (TNR) obtained from this experiment were used to compare discrimination performance. To compare the performance of each machine learning method and conventional method, the following indices are used as evaluation measures.

**True Positive Rate (TPR)**

The probability of correctly identifying a URL string labeled "attack" as "attack".

**True Negative Rate (TNR)**

The probability of correctly identifying a URL string labeled "clean" as "clean".

A discrimination method with a higher TPR (lower false negative rate) is suitable for this research because a higher True Positive Rate indicates that more URL strings that should be discriminated as "attack" were discriminated as "attack". On the other hand, a higher true negative rate (lower false positive rate) indicates that more URL strings that should be identified as "clean" are identified as "clean", and a discrimination method with a higher true negative rate is less likely to incorrectly identify the URL string "clean" URL string as "attack".

The True Positive Rate (TPR) and True Negative Rate (TNR) used as evaluation measures can be derived using the following equations.

$$TPR = \frac{TP}{TP + FN} \tag{1}$$

$$TNR = \frac{TN}{TN + FP} \tag{2}$$

In the above equation, $TP$ denotes a true positive (an attack was correctly judged as an attack), $FN$ a false negative (an attack was incorrectly judged as not an attack), $TN$ a true negative (a non-attack was correctly judged as not an attack), and $FP$ a false positive (a non-attack was incorrectly judged as an attack). (false positives).

Next, we describe the training and test data. The training data used were 33,088 accesses collected in the WordPress data collection environment over a three-month period from 29 October 2021, to 22 January 2022. The data used as test data consisted of 7195 ac-

cesses collected in the WOWHoneypot data collection environment during a one-month period from December 2021 to January 2022.

Next, the rules generator was used to generate regex patterns for Log4Shell's obfuscated attack patterns, which were converted into rules that could be applied to Snort, the IDS. We compared these rules with the rules included in the Snort Community ruleset. The evaluation metrics used in the comparison are described below.

**Detection rate**

The value of how many alerts were correctly issued for accesses that should have been alerted as attacks. The detection rate $DR$ can be derived by the following equation, where $TP$ is defined as true positive (alerts were issued for attacks) and $FN$ as false negative (no alerts were issued for attacks).

$$DR = \frac{TP}{TP + FN} \tag{3}$$

**Understandability**

This value represents the understandability of the regex pattern. Assuming that selection, repetition, negation, etc., are defined as complex regexes, with the number of complex regexes in the conventional method defined as $Complexity_{Conventional}$ and the number of complex regexes in the proposed method as $Complexity_{Proposed}$, the understandability is obtained by the following equation.

$$Understandability = 100 - \frac{Complexity_{Proposed}}{Complexity_{Conventional}} * 100 \tag{4}$$

Since this metric compares the number of complex regexes, the experiment examines the number of complex regexes in each regex pattern to calculate understandability.

The URL strings used to test the generated rules were generated for 161 accesses related to the Log4Shell exploit, out of approximately 250,000 accesses collected in the WordPress data collection environment from 29 October 2021 to 5 June 2023, for approximately 18 months. Rule generation was performed. The Snort Community ruleset was used for comparison with the generated rules, and one of the rules in the Snort Community ruleset was selected for performance comparison, corresponding to one of the attack patterns, including obfuscated ones that exploit Log4Shell. The 161 attack patterns used for rule generation include all the obfuscation methods of the attack patterns that exploit Log4Shell shown in Figure 1, as well as some attack patterns that are not obfuscated. Therefore, those attack patterns cover all variations of the attack patterns that exploit Log4Shell.

The prior knowledge used to generate the regex patterns and rules is shown below. Note that this prior knowledge is a modification of the prior knowledge shown in the example used to illustrate the rule generation flow so that it can handle uppercase alphabetic characters.

- `$\{upper:([a-zA-Z0-9]|:|:|\. |/)\}` is used as one of the obfuscation methods. Therefore, this obfuscation is called `upper obfuscation`.
- `$\{lower:([a-zA-Z0-9]|:|\. |/)\}` is used as one of the obfuscation methods. Therefore, this obfuscation is called `lower obfuscation`.
- `$\{([a-zA-Z0-9]|:)+:\-([a-zA-Z0-9])+\}` is used as one of the obfuscation methods. Therefore, this obfuscation is called `obfuscation of random strings`.
- If multiple obfuscation methods are used in combination, their order of appearance is not considered.

The attack patterns used to generate the regex patterns and rules are the 10 attack patterns generated by the program that generates the attack patterns for Log4Shell, which we created [35]. These variations and the number of each are shown in Table 2.

**Table 2.** Variations in attack patterns used for rule generation.

| Variations | Number of Each Variation |
|:---:|:---:|
| No obfuscation | 3 |
| upper or `lower` | 4 |
| random strings | 3 |

*4.2. Experimental Results*

A comparison between the conventional and proposed methods for the true positive and true negative rates for known attacks is shown in Figure 13. The y-axis is the value of each evaluation index, with higher values indicating better discrimination performance.

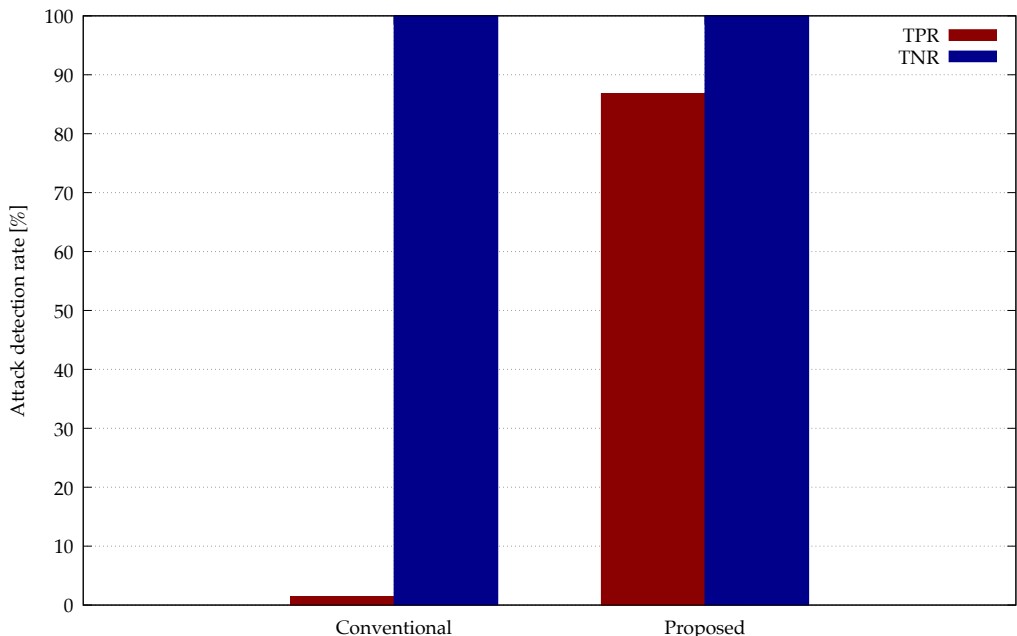

**Figure 13.** Comparison of TPR and TNR for known attacks.

Next, a comparison between the conventional and proposed methods for the true positive rate for variant attacks is shown in Figure 14. The y-axis is the value of the true positive rate, with higher values indicating more attacks can be detected.

Using the 10 attack patterns described above, rule generation resulted in one regex pattern shown below.

```
($\{([a-zA-Z0-9]|:)+:\-([a-zA-Z0-9]|:)+\}|[a-zA-Z]|[0-9]|$|\{|:|\-|/|\.|\})
+
```

The regex pattern shown above was then used to generate rules applicable to Snort. The generated rule is shown below.

```
alert tcp $EXTERNAL_NET any -> $HOME_NET $HTTP_PORTS (msg:''Log4Shell'';
flow:to_server,established; pcre:''/(\$\{([a-zA-Z0-9]|:)+:\-([a-zA-Z0-9]|:)
+\}|[a-zA-Z]|[0-9]|\$|\{|:|\-|/|\.|\})+/U''; metadata:policy balanced-ips
drop, policy connectivity-ips drop, policy max-detect-ips drop, policy
security-ips drop, ruleset community, service http; classtype:attempted-
user; sid:58724; rev:6;)
```

The results of applying the above rule to Snort and examining the attack detection rate using the aforementioned 161 attack patterns and comparing it with the conventional method using the Snort Community ruleset are shown in Figure 15. The y-axis is the attack detection rate, and a higher value indicates that the attack was detected.

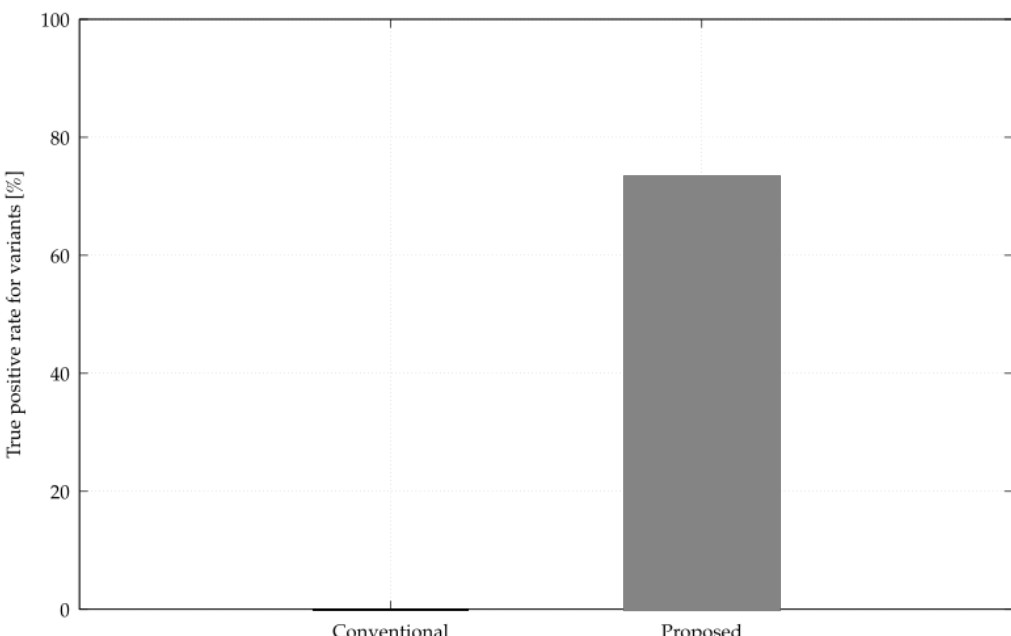

**Figure 14.** Comparison of TPR for variant attacks.

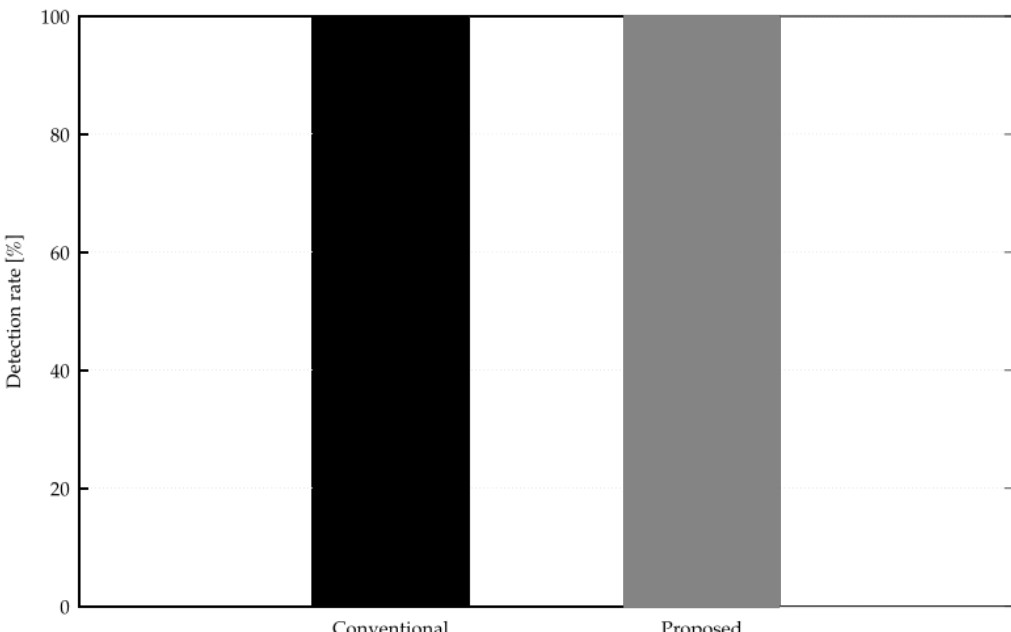

**Figure 15.** Detection performance comparison of rules generated by conventional and proposed methods.

Next, the comparison of the number of complex regex patterns of the rules generated by the conventional method and the proposed method is shown in Figure 16. The y-axis is the number of complex regexes, with lower values indicating greater understandability, as defined in the evaluation metrics.

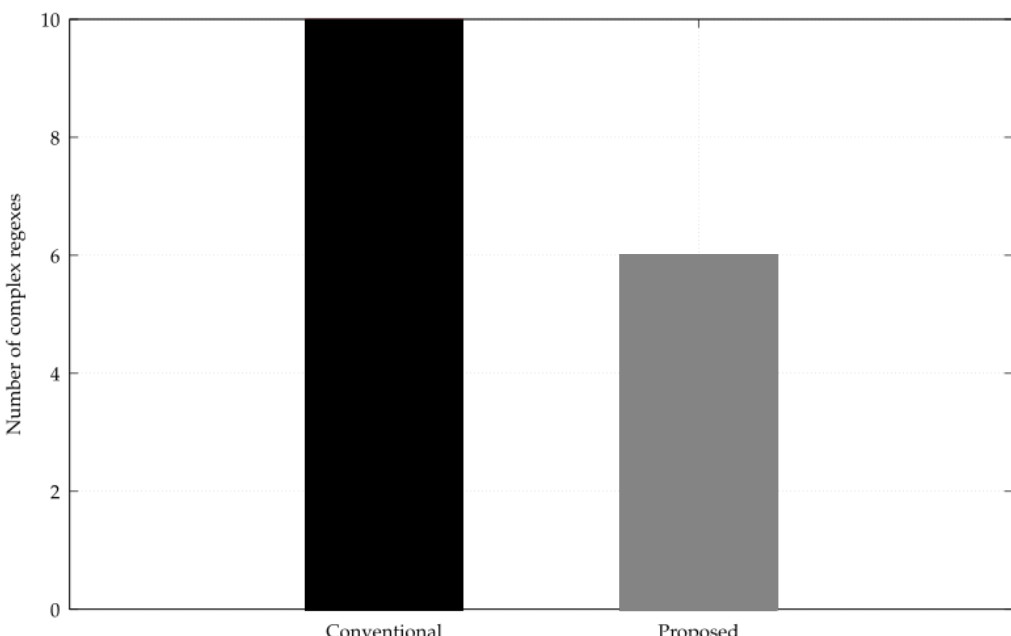

**Figure 16.** Comparison of the number of complex patterns of rules generated by conventional and proposed methods.

## 5. Discussion of Experimental Results

### 5.1. Discrimination of Attacks

First, we discuss the experimental results of discriminating between attacks and non-attacks. From Figure 13, it was found that the proposed method was able to correctly detect known attacks about 80% higher than the conventional method. On the other hand, the true negative rates were both 100% and there was no difference between the proposed and conventional methods. Unlike the true positive rate, the true negative rate is an evaluation metric of discrimination performance for non-attacks, not for attacks. Needless to say, since the detection of attacks is important, it is better to have a high true positive rate, which is an evaluation measure for attacks. However, in this paper, we added the true negative rate as one of the evaluation indices in addition to the true positive rate. This is because a method with a low TNR is likely to incorrectly judge accesses that are not originally attacks as attacks, resulting in repeated blocked accesses. But, since the proposed method has a TNR of 100%, non-attack accesses will not be unnecessarily blocked. Therefore, the proposed method shows high discrimination performance for both detection of known attacks and discrimination against non-attack accesses and is superior to the conventional method for known attacks.

Next, from Figure 14, it was found that the proposed method was able to correctly detect variant attacks about 70% higher than the conventional method. Compared to the detection rate of attacks against known attacks, the discrimination performance is about 10% lower but still more than 70%, and the proposed method is superior to the conventional method even for attacks against variants. Therefore, the proposed method has better discrimination performance for both known attacks and variant attacks compared to the conventional method.

### 5.2. Rules Generation

Next, we discuss the results of the rule generation experiments. One regex pattern was generated, and is once again shown below.

```
($\{([a-zA-Z0-9]|:)+:\-([a-zA-Z0-9]|:)+\}|[a-zA-Z]|[0-9]|$|\{|:|\-|/|\.|\})
+
```

The attack pattern that was first read when generating the regex pattern is shown below.

```
${${VWegfRIPvF:-jndi:ldap://127.0.0.1:1389/tes}${iTZDZZM:gS:-t}}
```

Of the obfuscation methods given as prior knowledge, the one used to generate the regex pattern was the obfuscation of random strings, and the part of `${iTZDZZM:gS:-t}` matched it. On the other hand, the part `${VWegfRIPvF:-jndi:ldap://127.0.0.1:1389/tes}` is obfuscation of random strings, but prior knowledge does not support symbols such as : and / and did not match either prior. To deal with these, the following prior knowledge was used.

```
[a-zA-Z]|[0-9]|$|\{|:|\-|/|\.|\}
```

There were no attack patterns processed afterward that could not be processed by the regex pattern generated by the first attack pattern loaded, and only one regex pattern was generated by processing 10 attack patterns, so no unnecessary regex patterns were generated. From these results, it can be said that the expected regex patterns were successfully generated for the attack patterns actually observed by the honeypot.

Next, Figure 15 shows the results of applying the rules applicable to IPS and IDS, which were created based on the generated regex patterns, to Snort to confirm whether attacks could be detected. The Snort Community ruleset, which is the conventional method, was also able to detect all attack patterns, confirming that its discrimination performance is equivalent to that of the conventional method. Next, a comparison of the number of complex regexes used in Figure 16 shows that the regex pattern generated by the proposed method is 40% higher understandability than that of the conventional method. Furthermore, the Snort Community ruleset does not have a clear rule generation process, whereas the proposed rule generation method performs pattern matching based on prior knowledge and generates regex patterns based on the results, so the rule generation process is clear. Therefore, the proposed method is superior to the conventional method in terms of explainability.

In addition, in checking the validity of the generated rules, we did not focus on false positives. This is because all the attack patterns used in the experiment are actually used as attack patterns, and those that are not attacks are not included. In addition, the 161 attack patterns used in the experiment include all obfuscation methods used in attack patterns that exploit Log4Shell, as well as attack patterns that are not obfuscated, covering all variations of Log4Shell's attack patterns. Considering that all of those 161 attack patterns can be discriminated against, we can say that potential false negatives do not occur.

In summary, these results show that both the attack discrimination performance and the effectiveness and understandability of the generated rule is superior to the compared conventional method and highly explainable.

The fact that we could respond to all of the observed attacks that exploit Log4Shell, even though we only provided 10 attack patterns, is important in a situation like Log4Shell where the interval between the vulnerability disclosure and the actual attack is short and the attack patterns change. The variation of attack patterns is a critical factor in the development of a system that can be used to detect and prevent attacks. Variations in attack patterns can be known at an early stage through programs that generate attack patterns and PoCs, but it is difficult to see how these variations are used in actual attacks and how they are combined until after many attacks have actually been observed. It is difficult to find out how they are used in actual attacks and how these variations are combined until many attacks are actually observed. However, the proposed method was able to handle attack patterns in which multiple obfuscation methods are combined or obfuscated and non-obfuscated portions are mixed. The fact that the proposed method can deal with attacks as early as possible before they become widespread, is effective in cases where attacks are launched immediately after the vulnerability is disclosed.

## 6. Conclusions

The method proposed in this paper has the following strengths:

**Modularized rule generation methods**

The proposed method can respond to attacks other than those that exploit Log4Shell by modifying or adding the prior knowledge it provides. Only prior knowledge is needed to deal with various attacks, and the rule generation method itself does not need to be modified.

**Generation of rules with high understandability**

In the proposed method, regex patterns and rules applicable to IPS and IDS are generated by combining obfuscation methods given as prior knowledge. By avoiding the use of complex regexes, rules with high understandability can be generated.

However, the method proposed in this paper has several limitations, which are described below.

**Handling of attacks other than Log4Shell**

Since only the obfuscation method of Log4Shell is given as prior knowledge, it is necessary to add prior knowledge for attacks other than Log4Shell. However, it is difficult to manually provide prior knowledge that covers all attack patterns, so a method that can automatically generate prior knowledge from information on the Web is needed.

**Simplification of generated regex patterns**

In the proposed method, the regex pattern of the obfuscation method given as prior knowledge is directly used for rule generation. However, there are cases where multiple obfuscation methods can be combined, as in the case of Log4Shell's obfuscation methods `upper` and `lower`, which have the `wer` part in common. If the length of the generated regex pattern as well as the use of complex regexes is important, it is also necessary to summarize obfuscation methods given as much prior knowledge as possible.

In the future, we intend to solve the above issues and be able to generate rules that can handle more attacks and have shorter lengths of the generated regex patterns.

**Author Contributions:** Conceptualization, Y.Y. and S.Y.; methodology, Y.Y.; software, Y.Y.; validation, Y.Y.; formal analysis, Y.Y.; investigation, Y.Y.; resources, Y.Y.; data curation, Y.Y.; writing—original draft preparation, Y.Y.; writing—review and editing, S.Y.; visualization, Y.Y.; supervision, S.Y.; project administration, S.Y.; funding acquisition, S.Y. All authors have read and agreed to the published version of the manuscript.

**Funding:** This research was funded by JSPS KAKENHI Grant Numbers JP19K11965 and JP22K12028.

**Data Availability Statement:** Data such as URL strings used in the experiments and attack patterns used in rule generation are available from https://yuudai-g.github.io/ (accessed on 10 July 2023).

**Conflicts of Interest:** The authors declare no conflict of interest.

## Abbreviations

The following abbreviations are used in this manuscript:

| | |
|---|---|
| IDS | Intrusion Detection System |
| IPS | Intrusion Prevention System |
| PoC(s) | Proof of Concept(s) |
| CVE | Common Vulnerabilities and Exposures |
| SVM | Support Vector Machine |
| TV | Text Vectorization |

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
