# Peer review of "Defense Mechanism to Generate IPS Rules from Honeypot Logs and Its Application to Log4Shell Attack and Its Variants"

_electronics, doi:10.3390/electronics12143177_

Round 1
Reviewer 1 Report
1. It is not entirely clear why, first of all, we are talking only about Log4Shell attacks, because the work considers an universal option, and Log4Shell is rather an example of application. At least, there are no features in the description of the method that indicate its focus on a specific attack, before the description of the experiment with data sets.
2. The statement that there are no methods that allow you to create autonomous defense systems or use black-boxed methods does not look proven. There is practically no review of similar methods and systems in the work. In particular, there is no clear definition of which method is considered conventional.
3. In general, the description does not make it possible to fully evaluate the scientific novelty of the proposed method, its advantages and limitations.
Reviewer 2 Report
See detailed comments in the attached file.

English proofreading is recommended.
Reviewer 3 Report
The paper proposes a defense system that can protect against Log4Shell Attack by generating IPS rules from Honeypot logs. The proposed system includes 3 parts: a honeypot to collect data, a machine learning module to analyze the collected data and identify attacks, and a module to generates rules that can be applied to an IPS. The paper is easy to understand, the problem being addressed is interesting. However, the novelty of the work is limited. Also, the paper contains several weaknesses that the authors should take into consideration:
- The section describing the system lacks many details concerning the model and its operating principle. The authors say they have used machine learning to analyze honeypot logs, but I was surprised by the lack of information about the algorithms used, how they are organized, how features are processed, and many other details that are missing in this section. Figure 4 is not very meaningful, and the authors should describe their model in more detail to be able to evaluate its effectiveness.
- The authors compared their model with a conventional method, but they did not provide any description of this method or the algorithms used. This greatly affects the credibility of the results presented in the paper.
- The authors have discussed the related works in the "Discussion of experimental results" section. In a research article, the related works should be described in detail in a separate section, generally before the description of the proposed system.
- A conclusion is missing at the end of the paper.
- Authors should improve the structure of the paper
- The current paper seems to be more of a technical report. The research aspect is missing and the novelty is very limited. In my opinion it is not possible to accept it as a research article in its present form.
Moderate editing of English language required
Reviewer 4 Report
1-Some acronyms are NOT defined in the paper such as CVE and PoCs.
2-More references are required in the introduction section such as for Suricata ET rule set and Snort Community rule set.
3-Authors said in line 165: “Machine learning is performed. The operator for performing machine learning can be changed to change the machine learning method used.” However, the operator to select the kind of the machine learning used by authors is NOT defined in the paper.
4-Authors said in line 168: “The same operators used for loading the training data are used to load the CSV file and change the data type. However, the CSV file structure used by authors is NOT given in the paper.
Reviewer 5 Report
(1) Main aim of the research:
The main aim of the research was to design an advanced method to detect and respond to the Log4Shell vulnerability, a major security flaw identified in the Apache Log4j library. The research developed a new technique that generates regex patterns and uses them to detect both known and variant attack patterns.
(2) Novelty of the research:
The research's novelty lies in its method of identifying and responding to the Log4Shell vulnerability. It proposes a novel regex pattern generation method that leverages prior knowledge and attack pattern obfuscation techniques. This approach appears to be more effective and accurate than conventional methods in detecting both known and variant attacks.
(3) Major contribution to the current research landscape:
The study contributes to the current research landscape by providing a novel and more effective method for detecting and mitigating Log4Shell vulnerability. This research not only improves upon the existing solutions but also contributes significantly to the body of knowledge about the handling of similar cyber security threats.
(4) Alignment with the aims and scope of the journal:
The paper is well aligned with the scope of the Electronics journal, specifically fitting into the areas of Computer Science & Engineering, Networks, Systems & Control Engineering, Circuit and Signal Processing, and Artificial Intelligence. It contributes new insights into cybersecurity, an issue of significant importance in the electronics domain.
(5) Major issues of the manuscript:
One major issue is the lack of detailed discussion on the limitations of the proposed method and how it compares to other existing methods in different scenarios. Also, a clear explanation about the potential false positives and negatives would have been beneficial.
(6) Internal validity:
The internal validity appears strong. The methodology was thoroughly detailed, the experiments were designed appropriately, and the results were analyzed and discussed comprehensively.
(7) External validity:
The external validity, while reasonable, is somewhat limited due to the authors' restricted citation base. There are 253 results of "Log4Shell" since 2019 on Google Scholar. To substantiate the universal applicability of their method, it would be beneficial for the authors to perform a more robust literature review. They should look into a wider range of sources, analyze and contrast their approach with existing solutions, and provide clear reasoning to validate why their proposed method is likely to yield superior outcomes.
(8) Are there any industry or commercial solutions that can already solve the issue that the authors are trying to address?
While there are commercial Intrusion Detection Systems (IDS) and Intrusion Prevention Systems (IPS) that can detect and respond to such vulnerabilities, the paper proposes a novel and potentially more effective solution to specifically address the Log4Shell vulnerability.
(9) Are the title, the abstract and the keywords appropriate and suitable for the contents?
Yes, the title, abstract, and keywords are all appropriate and clearly describe the content of the paper. They provide a good overview of the research and its findings.
(10) Comment on the level of English language used:
The English language usage in the article is of a high standard. It is well-structured, clear, and easy to understand. However, some sections could benefit from simpler language or further explanation to be more accessible to a broad readership.
(11) Suggestions for improvements of the paper:
- Extend the discussion to detail the limitations of the proposed method.
- Evaluate and contrast the proposed method's performance with other existing methods across various scenarios.
- Provide explicit information about the potential for false positives and negatives in the application of the proposed method.
- Improve visual clarity of the graphs by adding shadows or differentiating elements for better black-and-white print readability.
- Construct a more comprehensive literature review, incorporating additional academic studies specifically addressing Log4Shell.
(12) Justifications to suggest an acceptance, a major revision, or a rejection:
In light of the above comments, a major revision is recommended for this paper. While the study presents valuable findings and a novel method, there are areas that need more thorough investigation and clarification, particularly around external validity and literature review. Furthermore, improvements in visual representation and discussion of potential false outcomes will bolster the paper's strength and readability. The authors are encouraged to undertake these revisions to enhance the paper's contribution to the field.
The English language usage in the article is of a high standard. It is well-structured, clear, and easy to understand. However, some sections could benefit from simpler language or further explanation to be more accessible to a broad readership.
Round 2
Reviewer 1 Report
All my comments were taken into account
Author Response
We greatly appreciate your careful reviewing. We carefully reviewed the entire paper and corrected it appropriately.
Reviewer 2 Report
It is recommended to proofread the article.
Proofread the article before publication.
Reviewer 3 Report
The authors have made sufficient efforts to revise the manuscript.
Minor editing of English language required
Author Response

(The authors gave the same response as above.)

Reviewer 4 Report
I encourage the authors to address to following concerns mostly on presentation.
1) The following sentence in line 253 is incomplete:
The process shown in Figure 6 shows the process when Fast Large Margin is used, which can be modified to use another machine learning meth
2) The following sentence in lines 281 and 282 need to be fixed:
Although only the obfuscation method used for obfuscation has been included in the prior knowledge, this will be changed so that strings that always appear at the beginning and end of the attack pattern can also be included in the prior knowledge.
3) Add the figure number in sentence in line 292:
The flow of generating rules by the method of generating rules using prior knowledge is shown below.
Minor editing of English language required.
Reviewer 5 Report
Acceptance recommended after the author's revision.
Author Response

(The authors gave the same response as above.)
